# Factors Associated with Major Errors on Death Certificates

**DOI:** 10.3390/healthcare10040726

**Published:** 2022-04-13

**Authors:** Sangyup Chung, Sun-Hyu Kim, Byeong-Ju Park, Soobeom Park

**Affiliations:** Department of Emergency Medicine, University of Ulsan College of Medicine, Ulsan University Hospital, Ulsan 44044, Korea; 0734990@uuh.ulsan.kr (S.C.); 0734988@uuh.ulsan.kr (B.-J.P.); 0734989@uuh.ulsan.kr (S.P.)

**Keywords:** death certificate, cause of death, major errors, cancer

## Abstract

The objective of this study was to investigate errors on death certificates and factors associated with the occurrence of major errors. A retrospective analysis was conducted for six months in 2020 at a university training hospital. Errors were judged as major and minor errors according to the contribution to the process of determining the cause of death. Death certificates were classified into two groups with major errors (ME group) and without major errors (non-ME group). General characteristics of the death certificates, the main cause of death (cancer, cardiovascular disease, cerebrovascular disease, digestive disease, respiratory disease, genitourinary disease, intentional self-harm, external causes, and other causes), the number of causes of deaths written on the death certificate, and major and minor errors were investigated. The ME group had 127 cases out of 548 death certificates. The number of causes of deaths written on the death certificates and the total number of errors were higher in the ME group than in the non-ME group. Cardiovascular disease, cerebrovascular disease, digestive disease, respiratory disease, external causes, and other diseases as causes of deaths had higher risks of major errors on death certificates than cancer as a cause of death. The group with cancer as a cause of death had the lowest incidence of major errors and fewer causes of deaths. To reduce major errors, continuous education and feedback are needed for those who are qualified to issue a death certificate.

## 1. Introduction

To accurately reflect the underlying cause of death (UCOD) of a patient based on the death certificate only, all relevant information should be recorded. The certifier should not intentionally select some conditions related to the death for entry and reject others. The World Health Assembly has established a common format with guidelines to improve the international accuracy and usefulness of death certificates [1]. Most death certificates were issued electronically following the guidelines in Korea.

Nevertheless, errors on death certificates are common worldwide. Korea is no exception [2,3,4]. Several studies have reported that almost 95% of the death certificates have one or more errors [5,6,7,8,9,10] and that more than half of death certificates have major errors [6,7,8,9,11,12,13,14]. When death certificates only have minor errors, such as a format error, there is no difficulty in guessing the cause of death. However, if major errors occur, the cause of death may be distorted, which can confuse the death certificate reviewer. Particularly, if there is an error in the manner of death such as an error in the judgment of a natural death or unnatural death due to an external cause, it can affect the bereaved family and cause legal and social controversies [15]. In addition, if the cause of death is incorrectly determined, it can lead to inaccurate and unreliable national death statistics, which can affect the direction of the national health policy. Therefore, reducing major death certificate errors is important from individual, social, and national perspectives.

Previously, most studies on death certificate errors have focused on general characteristics and the frequency of errors [6,7,8,9,10,11,12,13,14,16,17,18,19,20]. However, studies on the characteristics of death certificates with major errors are limited [11]. Therefore, the objective of this study was to investigate the characteristics of major errors on death certificates and factors associated with the occurrence of major errors by comparing two groups of death certificates (with and without major errors) issued in one hospital. Death certificate errors were also compared between a group with a higher incidence of major errors and a group with a lower incidence of major errors.

## 2. Materials and Methods

This study performed a retrospective analysis of death certificates issued at a university training hospital located in South Korea from January 2020 to June 2020. The study hospital was the only university training hospital in a metropolitan city with 1.1 million people. Therefore, more severely ill patients were admitted in the study hospital than in other hospitals of the region. It was approved by the relevant Institutional Review Board. The judgment of errors was first made by three residents in the Department of Emergency Medicine who had received an education on issuing a death certificate, who had experience in issuing a death certificate, and who were currently training in the emergency room. A senior emergency physician who had previous research experience with death certificate errors conducted a secondary review of the judgment of errors. In the case of an ambiguous judgment even after the second review, a final consensus process with four people was carried out. All available medical records, laboratory tests, and radiologic tests were reviewed. A total of 553 death certificates were issued by the hospital over six months (from January 2020 to June 2020). Among them, there were five cases in which the UCOD was unknown or it was difficult to judge the error in the manner of death after the final consensus process. These cases were excluded. Postmortem examination reports of 11 cases during the study period were also excluded. The final error analysis was performed for 548 cases.

### Definition of Errors

Errors on death certificates were classified into major errors and minor errors according to the contribution to the process of determining the UCOD (Table 1) [6,7,8,9,10,11,13,14,16,17,19,20,21,22,23,24,25].

Major errors were defined as errors that affected the determination of the UCOD, including the following: (1) mode of death such as cardiac arrest, heart failure, or respiratory failure as the UCOD; (2) secondary conditions such as sepsis and esophageal varix bleeding as the UCOD without an antecedent cause of death; (3) ill-defined conditions such as cachexia, senility, symptoms or signs, or abnormal clinical or laboratory findings not elsewhere classified corresponding to ICD-10 codes for R00-R94 and R96-R99 as the UCOD; (4) improper sequence of time between the causes of death, such as aspiration pneumonia for UCOD and cerebral infarction for cause of death; (5) incompatible causal relationships, such as esophageal varix bleeding and cerebral infarction; (6) more than one cause of death on a single line; (7) incorrect manner of death; (8) unacceptable cause of death without evidence of an logical decision, such as pneumonia as the cause of death even though it should have been myocardial infarction; and (9) early-stage cancer as the UCOD in a cancer patient, regardless of the stage.

Minor errors were defined as errors that did not directly affect the determination of the UCOD, including the following: (1) mode of death as the cause of death with an appropriate UCOD; (2) a blank line between the causes of death or duplication of the same cause of death; (3) no record for the date of onset; (4) incorrect record for the date of onset; (5) incorrect record for the time of death; (6) incorrect or no record of the time interval; (7) incorrect or no record of other significant conditions; (8) abbreviations, typographic errors, and unnecessary comments; (9) incorrect or no record of the place of death; (10) no record of detailed information for poisoning; (11) no record of poisoning as the UCOD, even if the recorded accident was classified with ICD-10 code V01-Y89; (12) no record of a surgical opinion; (13) no record for the date of surgery; (14) incorrect classification or no records for the type of accident; (15) incorrect or no records for the intention of the accident; (16) incorrect or no record for the time of the accident; (17) incorrect or no record for the place of the accident; (18) incorrect or no record of the trauma mechanism; and (19) no record of the cause of death, even if the trauma mechanism was recorded.

As for general characteristics on death certificates, the basic information of the patient present on the death certificate was investigated. The number of causes of deaths listed on the death certificate, the analysis of each detailed error, the number of major and minor errors, the total number of errors, the manner of death (whether the UCOD was a natural death or unnatural death due to an external cause), the place where the death certificate was issued, the department that issued the death certificate, and the main UCODs were also investigated. The place where the death certificate was issued was classified into three categories: the emergency department, intensive care unit, and general ward. The department that issued the death certificate was classified into nine categories (hemato–oncology, emergency medicine, pulmonary, gastroenterology, neurosurgery, nephrology, general surgery, cardiology, and other departments) in the order of the number of issuances. The main UCOD was classified into nine categories (cancer, cardiovascular disease, cerebrovascular disease, digestive disease, respiratory disease, genitourinary disease, intentional self-harm, external causes, and other causes) based on systematic classification with ICD-10 codes. Circulatory diseases and diseases by an external cause of morbidity were subdivided into cardiovascular disease, cerebrovascular disease, and intentional self-harm, and external causes according to the death rank in 2019 published by Statistics Korea [26]. Pneumonia was included in the category of respiratory disease considering its low number of death certificates. Causes of deaths with the number of death certificates not more than 10 were grouped into the category of other causes.

Death certificates were classified into groups with major errors (ME group) and groups without major errors (non-ME group). The Chi-squared test and Fisher’s exact test for categorical variables and the Mann–Whitney U test for non-normally distributed continuous variables were used to analyze the general characteristics and frequency of death certificate errors. To identify factors related to the occurrence of major death certificate errors, a univariate logistic regression analysis of general characteristics of death certificates was first performed to identify variables with *p* < 0.05. A multivariate logistic regression was then performed using significant factors identified in the univariate analysis. Among different variables, frequent UCODs and departments that issued the death certificate were highly correlated. Thus, only the frequent UCODs were included in the univariate logistic regression. To compare characteristics of death certificates according to the UCOD, death certificates were classified into a cancer group and a non-cancer group, since cancer was the leading UCODs on death certificates with fewer major errors. All statistical analyses were performed with IBM SPSS version 24.0 (IBM, Armonk, NY, USA). Statistical significance was defined as *p* < 0.05.

## 3. Results

General characteristics of death certificates and frequencies of errors are shown in Table 2. Of the 548 death certificates, 127 (23.2%) were in the ME group and 421 (76.8%) were in the non-ME group. The median age was 68 years in the ME group and 65 years in non-ME group. Neither age nor gender was significantly different between the two groups. The most common place of death certificate issuance was the intensive care unit with 53 (41.7%) cases in the ME group and the general ward with 252 (61.3%) cases (*p* < 0.001) in the non-ME group. The most common department of issuance was the department of pulmonology (26.0%) followed by the department of emergency medicine (24.4%) in the ME group, whereas it was the department of hemato–oncology (45.1%) followed by the department of emergency medicine (14.3%) and the department of pulmonology (12.6%) in the non-ME group. In the case of the manner of death, 9.4% of deaths in the ME group and 10.0% of deaths in the non-ME group were due to external causes. There was no significant difference in the number of lines filled for the cause of death. In both groups, most had only one line filled for the cause of death. The number of total death certificate errors was higher than the non-ME group (*p* < 0.001), but there was no difference in the number of minor errors. The main underlying causes of death were other diseases (23.6%) and cardiovascular disease (15.7%) in the ME group. However, cancer (64.4%) was the most common UCOD in the non-ME group (Table 2).

Place of issuance, department of issuance, and UCOD that were significant factors for the occurrence of major errors in the univariate analysis were used as covariates for the multivariate analysis. Issuing a death certificate in the emergency department had a higher risk for major errors than that in the general ward. The departments of emergency medicine, pulmonology, nephrology, cardiology as the issuing departments had higher risks for major errors than the Department of hemato–oncology. The category of cardiovascular disease, cerebrovascular disease, digestive disease, respiratory disease, external causes, and other diseases as the UCOD had a higher risk for major errors than cancer as the UCOD (Table 3).

There were 290 (52.9%) cases in which cancer was the leading UCOD. They were assigned to the cancer group. There were 258 (47.1%) cases with other underlying causes of deaths. They were assigned to the non-cancer group. The most common place of issuance was the general ward (230 cases, 79.3%) in the cancer group and the intensive care unit (130 cases, 50.4%) in the non-cancer group. The most common department of issuance was the department of hemato–oncology (198 cases, 68.3%) in the cancer group and the department of emergency medicine (69 cases, 26.7%) in the non-cancer group. The median number of lines filled for the cause of death was 1.3 in the cancer group, which was less than 1.8 in the non-cancer group (*p* < 0.001). More than one cause of death was listed on 22.8% and 51.9% of death certificates in the cancer group and the non-cancer group, respectively. The number of major errors was 0.2 in the cancer group, which was less than that (0.5) in the non-cancer group (*p* < 0.001) (Table 4).

Categories with fewer major errors in the cancer group than in the non-cancer group included the mode of death, secondary conditions, and ill-defined conditions as underlying causes of death, incompatible causal relationships, more than one cause of death on a single line, an incorrect manner of death, and an unacceptable cause of death (Table 5).

## 4. Discussion

Previous studies have reported that major errors occur in more than half of death certificates and that minor errors occur in most death certificates. However, only 23.2% (127/548) of death certificates in this study had major errors. The reason might be because cancer as the UCOD was high at 52.9% (290/548) in the present study, unlike previous studies.

Cancer has characteristics that are comparable to those of other diseases. If cancer (whether primary or metastatic) is the focus of care during a relevant episode of health care, it should be recorded and coded as the “main condition” [1]. That is, if cancer is listed as the UCOD, a major error related to the determination of the UCOD will not occur. Thus, there is no need to worry about it when issuing a death certificate. Considering that the most common major error that can affect the determination of the UCOD is an unacceptable cause of death [6,7,12], the possibility of a major error occurring by listing another inappropriate state on the death certificate is significantly reduced, since the diagnosis is certain because cancer is confirmed in patients by radiologic tests and biopsies during hospitalization, and the disease code is registered. In contrast, minor errors occurred similarly on all death certificates regardless of whether or not a major error occurred (Table 2). Similar results were seen regardless of whether the group was classified into a cancer group or a non-cancer group (Table 4). This meant that, in most cases, minor errors occurred regardless of the UCOD, consistent with previous studies [7,9,14,16].

There was a significant difference in the occurrence of major errors according to the place of death certificate issuance (Table 3). This seems to be because it is relatively common that a patient in the emergency room deceases before the examination is completed and a diagnosis is made, compared to a patient in a general ward. Especially, cancer patients usually have outpatient or hospitalization records after their diagnosis, making it easy to determine the UCOD, regardless of the place of issuance of the death certificate, which can reduce the possibility of major errors related to the determination of the UCOD.

Before the start of this study, it was thought that there would be a difference in the occurrence of major errors according to the number of causes of deaths listed on the death certificate. However, the multivariate analysis did not show statistically significant results (Table 3). This might be because major errors often occurred by listing only the mode of death without another UCOD, such as listing heart failure as the only cause of death or listing obviously secondary conditions as the UCOD without an antecedent cause of death, such as pneumonia as the only cause of death when aspiration pneumonia secondarily occurred in a cerebral infarct patient. However, when the UCOD was cancer, the number of causes of deaths listed on the death certificate was less than that for other diseases (Table 4). We found that major errors were significantly reduced when cancer was the only cause of death listed on the death certificate, regardless of the mode of death. Especially, only 5.5% (11/201) of major errors occurred in the department of hemato–oncology, where most underlying causes of deaths were cancer. That is, if the diagnosis was certain, there was no difficulty in determining the UCOD, even if fewer causes of deaths were listed. Besides, the possibility of major errors occurring on the death certificate was reduced.

Considering all these points, he findings of this study support the argument that death certificate errors can be reduced and that the quality of death certificates is improved when the UCOD is cancer [27,28]. Thus, it can be judged that the accuracy of death statistics for cancer is higher than that for other disease groups that are underestimated/overestimated in the death statistics [29,30]. As the occurrence of death certificate errors highly affects the death statistics [6,7,11,31], if a change in the mortality rate for cancer is accurately understood, it will give strength to the direction of cancer-related medical policies [12].

We found some cases with multiple causes of deaths listed on death certificates, in which major errors did not occur even if the UCOD was not cancer because it was easy to follow up by detailed progress notes during long-term hospitalization. On the other hand, we found some cases in which major errors occurred by listing only the mode of death such as cardiac arrest and unspecified (ICD-10 code I46.9) as the UCOD or listing only secondary conditions such as pneumonia and unspecified (ICD-10 code J18.9) as the UCOD. Despite the fact that the death certificate should have few errors, there is also a lack of awareness of the importance of death certificates. Major errors are inevitable due to the lack of verification systems and feedback after issuance of a death certificate [19]. Many previous studies have suggested that educational intervention is necessary to reduce death certificate errors [8,9,14,16,18,19,20]. Results of this study indicate the necessity of education on how to simplify the death certificate and accurately list the UCOD without missing it. In addition, feedback is needed not only for certifiers in the emergency department, but also for certifiers in the general wards and intensive care units.

This study had several limitations. First, this study targeted only death certificates issued in one training hospital. Thus, there is a limitation in generalizing the results of this study. Considering that distribution of the UCOD is highly variable depending on the issuing institution, in the case of death certificates issued by higher medical institutions, the proportion of cancer listed as the UCOD was high. Thus, the distribution of UCODs should vary when interpreting the results of this study. Second, there might have been differences in the education on death certificate issuance, experience with death certificate issuance, and clinical experience depending on the certifier of the death certificate. However, we did not investigate differences in the certifiers in this study. Third, some studies have been conducted on the presence of death certificate errors for specific cancers [32,33]. However, we did not investigate death certificate error differences by the type of cancer in this study because the number of subjects highly varied depending on the type of cancer within the limited number of death certificates. The period and number of death certificates should be extended to investigate differences by the type of cancer. Fourth, a previous study considered that listing cancer as the only cause of death was an error [30]. Among death certificates where the UCOD was cancer, cancer was listed as the only cause of death on 77.2% (224/290) of death certificates. If cancer was the only cause of death listed on the death certificate, the main condition remains unchanged although there might be a lack of detailed information for the cause of death. This does not affect the death statistics. Therefore, from individual and social perspectives, it could be considered that major errors related to the determination of the UCOD did not occur. Fifth, major error number 6 “More than one cause of death on a single line” can be thought to be a minor error, because the WHO guideline has stated that if more than one causes are reported on a single line, the first cause should be selected unless there is a cause-and-effect relationship for the reported causes. If the first cause of multiple causes on a single line was the most appropriate as the cause of death, that error would not be considered as a major error. However, if the first cause was selected as the cause of death, the acceptable cause of death had only 3 (23%) cases of 13 cases in this study. Therefore, the error with more than one cause of death on a single line was classified as a major error in this study.

## 5. Conclusions

Of all death certificates in this study, 23.2% had major errors. When cancer was listed as the UCOD, there was a lower risk of major errors occurring on the death certificate compared to when the UCOD was a cardiovascular disease, cerebrovascular disease, digestive disease, respiratory disease, external causes, or other diseases, considering the clarity of a diagnosis and accessibility of medical records. It is important to understand the main characteristics of death certificate errors. Continuous education and feedback are needed for those who are qualified to issue a death certificate to reduce major errors.

## Figures and Tables

**Table 1 healthcare-10-00726-t001:** Definitions of major and minor errors on death certificates.

Type of Error	Definition
Major errors	
(1) Mode of death as the UCOD	Listing only the mode of death without another UCOD
(2) Secondary condition as the UCOD	Listing obviously secondary conditions as the UCOD without an antecedent cause of death
(3) Ill-defined conditions as the UCOD	Listing only ill-defined conditions as the UCOD
(4) Improper sequence	Indicating an improper sequence of time between the causes of death
(5) Incompatible causal relationship	Listing an incompatible causal relationship
(6) More than one cause of death on a single line (7) Incorrect manner of death	Listing more than one cause of death on a single lineIndicating a wrong judgment for the manner of death, such as a natural cause or an external cause
(8) Unacceptable cause of death	Listing an unacceptable cause of death without evidence of a logical decision
(9) Early-stage cancer as the UCOD *	Listing early-stage cancer as the UCOD, regardless of the stage
Minor errors	
(1) Mode of death as the cause of death with an appropriate UCOD	Listing the mode of death as the cause of death, even if an appropriate UCOD is included
(2) Blank/duplication ^†^	Including a blank line between the causes of death or a duplication of the same cause of death
(3) No record for the date of onset	No record for the date of onset
(4) Incorrect date of onset	Listing an incorrect record for the date of onset
(5) Incorrect time of death	Listing an incorrect record for the time of death
(6) Incorrect time interval	Listing incorrect or no records of the time interval
(7) Incorrect other significant conditions	Listing incorrect or no records of other significant conditions
(8) Abbreviation/typographical error/unnecessary comment	Listing an abbreviation, typographical error, or an unnecessary comment
(9) Incorrect place of death	Listing incorrect or no records for the place of death
(10) No record of detailed information for poisoning ^‡^	No record of detailed information for poisoning
(11) No record of poisoning as the UCOD ^‡^	No record of poisoning as the UCOD, even if recorded as the type of accident
(12) No record of a surgical opinion	No record of a surgical opinion
(13) No record for the date of surgery	No record for the date of surgery
(14) Incorrect type of accident	Listing an incorrect classification or no records for the type of accident
(15) Incorrect intention of the external cause	Listing incorrect or no records for the intention of the accident
(16) Incorrect time of an accident	Listing incorrect or no records for the time of an accident
(17) Incorrect place of an accident	Listing incorrect or no records for the place of an accident
(18) No record of the trauma mechanism ^§^	Listing only the cause of death without other trauma mechanisms
(19) Record of the trauma mechanism without another cause of death ^§^	Listing only the trauma mechanism without another cause of death

* Reference numbers are [20,21]; ^†^ Reference numbers are [16,22]; ^‡^ Reference number is [19]; ^§^ Reference number is [18]; Reference numbers for all other definitions without superscripts are [3,4,5,6,7,8,10,11,13,14,17]; UCOD, underlying cause of death.

**Table 2 healthcare-10-00726-t002:** Characteristics of death certificates according to major errors.

Characteristics	Major Error (*n* = 127)	Non-Major Error (*n* = 421)	*p*-Value
Age, years	68.0 (49.7–86.3)	65.0 (49.3–80.4)	0.057
Sex, male (%)	70 (55.1)	258 (61.3)	0.214
Place of issuance, *n* (%)			0.000
General ward	30 (23.6)	252 (59.9)	
Intensive care unit	53 (41.7)	110 (26.1)	
Emergency department	44 (34.6)	59 (14.0)	
Department of issuance			0.000
Hemato–oncology	11 (8.7)	190 (45.1)	
Emergency medicine	31 (24.4)	60 (14.3)	
Pulmonology	33 (26.0)	53 (12.6)	
Gastroenterology	11 (8.7)	42 (10.0)	
Neurosurgery	6 (4.7)	22 (5.2)	
Nephrology	11 (8.7)	15 (3.6)	
General surgery	7 (5.5)	11 (2.6)	
Cardiology	9 (7.1)	6 (1.4)	
Others	8 (6.3)	22 (5.2)	
Manner of death			0.861
Natural death	115 (90.6)	379 (90.0)	
External cause	12 (9.4)	42 (10.0)	
Number of lines filled for the cause of death	1.6 (0.7–2.5)	1.5 (0.7–2.3)	0.091
One, *n* (%)	74 (58.3)	274 (65.1)	
Two, *n* (%)	34 (8.1)	100 (23.8)	
Three, *n* (%)	12 (9.4)	36 (8.6)	
Four, *n* (%)	7 (5.5)	11 (2.6)	
Total number of errors on the death certificate	4.0 (1.9–6.1)	2.7 (1.5–3.9)	0.000
Number of major errors	1.1 (0.7–1.5)		
Number of minor errors	2.9 (1.0–4.8)	2.7 (1.5–3.9)	0.267
UCOD			0.000
Cancer	19 (15.0)	271 (64.4)	
Cardiovascular disease	20 (15.7)	22 (5.2)	
Cerebrovascular disease	6 (4.7)	16 (3.8)	
Digestive disease	14 (11.0)	25 (5.9)	
Respiratory disease	13 (10.2)	24 (5.7)	
Genitourinary disease	5 (3.9)	10 (2.4)	
Intentional self-harm	2 (1.6)	15 (3.6)	
External causes	18 (14.2)	27 (6.4)	
Others	30 (23.6)	11 (2.6)	

Numerical variables are presented as the median (interquartile range); UCOD, underlying cause of death.

**Table 3 healthcare-10-00726-t003:** Factors associated with major errors on death certificates.

	Univariate Logistic Regression			Multivariate Logistic Regression		
	Odds Ratio	Confidence Interval	*p*-Value	Odds Ratio	Confidence Interval	*p*-Value
Place of issuance						
General ward	1			1		
Intensive care unit	4.047	2.453~6.677	0.000	1.278	0.646~2.531	0.481
Emergency department	6.264	3.637~10.791	0.000	67.996	11.865~389.666	0.000
Department of issuance						
Hemato–oncology	1			1		
Emergency medicine	8.924	4.230~18.828	0.000	0.061	0.010~0.375	0.003
Pulmonology	10.755	5.094~22.704	0.000	5.491	1.803~16.727	0.003
Gastroenterology	4.524	1.839~11.128	0.001	1.770	0.462~6.782	0.405
Neurosurgery	4.711	1.587~13.987	0.005	1.030	0.207~5.126	0.971
Nephrology	12.667	4.719~33.997	0.000	6.246	1.534~25.436	0.011
General surgery	10.992	3.566~33.884	0.000	2.414	0.489~11.932	0.280
Cardiology	25.909	7.815~85.896	0.000	5.911	1.167~29.937	0.032
Others	6.281	2.283~17.280	0.000	1.860	0.456~7.587	0.387
UCOD						
Cancer	1			1		
Cardiovascular disease	12.967	6.042~27.825	0.000	5.706	2.101~15.501	0.001
Cerebrovascular disease	5.349	1.877~15.242	0.002	4.367	1.165~16.364	0.029
Digestive disease	7.987	3.579~17.825	0.000	5.148	1.627~16.288	0.005
Respiratory disease	7.726	3.404~17.536	0.000	2.910	1.027~8.247	0.044
Genitourinary disease	7.132	2.214~22.977	0.001	2.823	0.673~11.851	0.156
Intentional self-harm	1.902	0.405~8.933	0.415	0.961	0.183~5.039	0.962
External causes	9.509	4.463~20.258	0.000	7.126	2.557~19.861	0.000
Others	38.900	16.914~89.463	0.000	24.100	9.146~63.504	0.000

UCOD, underlying cause of death.

**Table 4 healthcare-10-00726-t004:** Characteristics of death certificates according to cancer diagnosis.

Characteristics	Cancer (*n* = 290)	Non-Cancer (*n* = 258)	*p*-Value
Age, years	66.2 (53.5–78.9)	65.0 (45.4–84.6)	0.384
Sex, male (%)	177 (61.0)	151 (58.5)	0.550
Place of issue, *n* (%)			0.000
General ward	230 (79.3)	52 (20.2)	
Intensive care unit	33 (11.4)	130 (50.4)	
Emergency department	27 (9.3)	76 (29.5)	
Department of issuance			0.000
Hemato–oncology	198 (68.3)	3 (1.2)	
Emergency medicine	22 (7.6)	69 (26.7)	
Pulmonology	27 (9.3)	59 (22.9)	
Gastroenterology	22 (7.6)	31 (12.0)	
Neurosurgery	1 (0.3)	27 (10.5)	
Nephrology	6 (2.1)	20 (7.8)	
General surgery	2 (0.7)	16 (6.2)	
Cardiology	1 (0.3)	14 (5.4)	
Others	11 (3.8)	19 (7.4)	
Manner of death			0.000
Natural death	290 (100)	204 (79.1)	
External cause	0 (0)	54 (20.9)	
Number of lines filled for the cause of death	1.3 (0.7–1.9)	1.8 (0.9–2.7)	0.000
One, *n* (%)	224 (77.2)	124 (48.1)	
Two, *n* (%)	50 (17.2)	84 (32.6)	
Three, *n* (%)	13 (4.5)	35 (13.6)	
Four, *n* (%)	3 (1.0)	15 (5.8)	
Total number of errors on the death certificate	2.7 (1.8–3.6)	3.3 (1.2–5.4)	0.000
Number of major errors	0.2 (0.0–0.4)	0.5 (0.0–1.1)	0.000
Number of minor errors	2.7 (3.5–1.9)	2.8 (0.9–4.7)	0.179

Numerical variables are presented as median (interquartile range).

**Table 5 healthcare-10-00726-t005:** Errors on death certificates according to a cancer diagnosis.

	Cancer (*n* = 290)	Non-Cancer (*n* = 258)	*p*-Value
Major errors, *n* (%)			
(1) Mode of death as the UCOD	0 (0)	21 (8.1)	0.000
(2) Secondary condition as the UCOD	5 (1.7)	48 (18.6)	0.000
(3) Ill-defined conditions as the UCOD	1 (0.3)	10 (3.9)	0.003
(4) Improper sequence	0 (0)	1 (0.4)	0.471
(5) Incompatible causal relationship	3 (1.0)	12 (4.7)	0.010
(6) More than one cause of death on a single line	3 (1.0)	10 (3.9)	0.029
(7) Incorrect manner of death	0 (0)	8 (3.1)	0.002
(8) Unacceptable cause of death	1 (0.3)	7 (2.7)	0.029
(9) Early-stage cancer as the UCOD	3 (1.0)	0 (0)	0.251
Minor errors, *n* (%)			
(1) Mode of death as the cause of death with an appropriate UCOD	11 (3.8)	41 (15.9)	0.000
(2) Blank/duplication	4 (1.4)	4 (1.6)	1.000
(3) No record for the date of onset	256 (88.3)	148 (57.4)	0.000
(4) Incorrect date of onset	7 (2.4)	11 (4.3)	0.225
(5) Incorrect time of death	0 (0)	0 (0)	
(6) Incorrect time interval	272 (93.8)	222 (86.0)	0.002
(7) Incorrect other significant conditions	209 (72.1)	155 (60.1)	0.003
(8) Abbreviation/typographical error/unnecessary comment	0 (0)	4 (1.6)	0.049
(9) Incorrect place of death	0 (0)	1 (0.4)	0.471
(10) No record of detailed information for poisoning	0 (0)	2 (0.8)	0.221
(11) No record of poisoning as the UCOD	0 (0)	0 (0)	
(12) No record of a surgical opinion	9 (3.1)	33 (12.8)	0.000
(13) No record for the date of surgery	8 (2.8)	21 (8.1)	0.005
(14) Incorrect type of accident	0 (0)	9 (3.5)	0.001
(15) Incorrect intention of the external cause	0 (0)	8 (3.1)	0.002
(16) Incorrect time of an accident	0 (0)	17 (6.6)	0.000
(17) Incorrect place of an accident	0 (0)	30 (11.6)	0.000
(18) No record of the trauma mechanism	0 (0)	31 (12.0)	0.000
(19) Record of the trauma mechanism without another cause of death	0 (0)	4 (1.6)	0.049

UCOD, underlying cause of death.

## Data Availability

Data and materials are available from the corresponding author upon reasonable request.

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
