# Peer review of "Factors Associated with Major Errors on Death Certificates"

_healthcare, 2022, doi:10.3390/healthcare10040726_

Round 1

Reviewer 1 Report

Abstract:

  1. In the abstract section, the time and place of the study should be stated. 
  2. It is suggested to briefly add appropriate solutions to improve the current situation.

Introduction:

  1. To prevent the repetition of the phrase "underlying cause of death" in all sections of the article and the existence of the well-known abbreviation UCOD of the “underlying cause of death” in articles related to death certificate errors, it is suggested that in the first use of the phrase "underlying cause of death" in the introductory section of the full form of the phrase with its equivalent abbreviation (i.e., UCOD), In other repetitions across the article, only its abbreviation should be used.
  2. Background on the process and how to complete a death certificate (whether manual or electronic) in Korea is essential in the introduction. 
  3. Because the word “studies” is used in the sentence, more than one reference is expected to be included for the sentence “However, studies on characteristics of death certificates with major errors are limited”. 

Methods:

  1. How many postmortem examination reports have there been in the same period? 
  2. In Table 1. It would be better to include the reference of the definitions of each error in front of each definition. 
  3. I believe that error number 6 “(6) More than one cause of death on a single line” in the major errors section is a minor error, not a major one. Because the World Health Organization guideline explicitly states that if there is more than one cause reported on a single line, the first cause should be selected unless there is a cause-and-effect relationship between the reported causes. What justification do the authors have for classifying this type of error as a major error? 
  4. The two paragraphs on page 3 after Table 1 are a repetition of the detailed definition of major and minor errors in Table 1. It is suggested that in order to prevent duplication of definitions, the two paragraphs after the table should be merged with the table definitions and only the table should be preserved and the next two paragraphs should be deleted. 

Results: 

The results section is written correctly and logically. Therefore, I have no comment on this section.

Discussion 

The authors discuss the findings of the study favorably. I have no comment in this section. 

Conclusion

In the conclusion section, it is suggested that the authors express the determinant factors of the occurrence of major errors in a sentence. 

References:

References need to be revised to include essential fields such as doi. 

Author Response

Thank you for your valuable time and comments. These comments have improved the quality of our manuscript significantly. We have revised this manuscript according to your comments or suggestions.

Reviewer 1

Comments and Suggestions for Authors

Abstract:

â–¶ In the abstract section, the time and place of the study should be stated. 

→ We have added the following statement in the abstract section.

Retrospective analysis was conducted for six months in 2020 at a university training hospital.

â–¶ It is suggested to briefly add appropriate solutions to improve the current situation.

→ We have added the following statement in the abstract section.

To reduce major errors, continuous education and feedback are needed for those who are qualified to issue a death certificate.

Introduction:

â–¶ To prevent the repetition of the phrase "underlying cause of death" in all sections of the article and the existence of the well-known abbreviation UCOD of the “underlying cause of death” in articles related to death certificate errors, it is suggested that in the first use of the phrase "underlying cause of death" in the introductory section of the full form of the phrase with its equivalent abbreviation (i.e., UCOD), In other repetitions across the article, only its abbreviation should be used.

→ As suggested, we have added the abbreviation for underlying cause of death (UCOD) and replaced all full names with this acronym to avoid repletion.  

â–¶ Background on the process and how to complete a death certificate (whether manual or electronic) in Korea is essential in the introduction. 

→ We thank the reviewer for pointing this out and we agree with the reviewer. Therefore, we have added the following statement in the introduction section.

Most death certificates were issued electronically following the guideline in Korea.   

Nevertheless, errors on death certificates are common worldwide. Korea is no exception [2-4].

â–¶ Because the word “studies” is used in the sentence, more than one reference is expected to be included for the sentence “However, studies on characteristics of death certificates with major errors are limited”. 

→ We have changed the word of “studies” to “study” since only one reference was cited.

Before

However, studies on characteristics of death certificates with major errors are limited

After

However, study on characteristics of death certificates with major errors is limited.

Methods:

â–¶ How many postmortem examination reports have there been in the same period? 

→ During the study period, there were 11 postmortem examination reports. We have changed the statement as following in the method section.

Before

Postmortem examination reports were also excluded.

After

Postmortem examination reports of eleven cases during the study period were also excluded.

â–¶ In Table 1. It would be better the include the reference of the definitions of each error in front of each definition. 

→ We have indicated the references for definition of each error in Table 1 as following.

Table 1. Definitions of major and minor errors on death certificates

Type of error

Definition

Major errors

Mode of death as the UCOD

Listing only the mode of death without another UCOD

Secondary condition as the UCOD

Listing obviously secondary conditions as the UCOD without an antecedent cause of death

Ill-defined conditions as the UCOD

Listing only ill-defined conditions as the UCOD

Improper sequence

Indicating an improper sequence of time between cause of deaths

Incompatible causal relationship

Listing an incompatible causal relationship

More than one cause of death on a single line

Incorrect manner of death

Listing more than one cause of death on a single line

Indicating a wrong judgment for the manner of death such as a natural cause or an external cause

Unacceptable cause of death

Listing an unacceptable cause of death without evidence of a logical decision

Early-stage cancer as the UCOD*

Listing early-stage cancer as the UCOD regardless of the stage

Minor errors

Mode of death as the cause of death with an appropriate UCOD

Listing the mode of death as the cause of death even if an appropriate UCOD is included

Blank/duplication

Including a blank line between cause of deaths or duplication of the same cause of death

No record for the date of onset

No record for the date of onset

Incorrect date of onset

Incorrect time of death

Incorrect time interval

Listing an incorrect record for the date of onset

Listing an incorrect record for the time of death

Listing incorrect or no records of the time interval

Incorrect other significant conditions

Listing incorrect or no records of other significant conditions

Abbreviation/typographical error/unnecessary comment

Listing an abbreviation, typographical error, or an unnecessary comment

Incorrect place of death

Listing incorrect or no records for the place of death

No record of detailed information for poisoning

No record of detailed information for poisoning

No record of poisoning as the UCOD

No record of poisoning as the UCOD even if recorded as the type of accident

No record of a surgical opinion

No record of a surgical opinion

No record for the date of surgery

No record for the date of surgery

Incorrect type of accident

Listing an incorrect classification or no records for the type of accident

Incorrect intention of the external cause

Listing incorrect or no records for the intention of the accident

Incorrect time of an accident

Listing incorrect or no records for the time of an accident

Incorrect place of an accident

Listing incorrect or no records for the place of an accident

No record of the trauma mechanism§

Listing only the cause of death without other trauma mechanisms

Record of trauma mechanism without another cause of death§

Listing only the trauma mechanism without another cause of death

*reference numbers are 20,21; reference numbers are 16,22; reference number is 19; §reference number is 18; reference numbers for all other definitions without superscripts are 3-8,10,11,13,14,17

â–¶ I believe that error number 6 “(6) More than one cause of death on a single line” in the major errors section is a minor error, not a major one. Because the World Health Organization guideline explicitly states that if there is more than one cause reported on a single line, the first cause should be selected unless there is a cause-and-effect relationship between the reported causes. What justification do the authors have for classifying this type of error as a major error? 

→ Thank you for your opinion. The number of major error (6), “more than one cause of death on a single line” was 13 in this study. Seven cases had a cause-and-effect relationship and three cases had no exact cause of death if the first cause was selected. Only 3 of the 13 cases had the error. “more than one cause of death on a single line” was correct if the first cause was selected as the cause of death. Therefore, cases with the error for “more than one cause of death on a single line” had a high risk for erroneously selection as the cause of death. We have added the above consideration in the discussion limitation section as follows.

Fifth, major error number 6 “More than one cause of death on a single line” can be thought to be a minor error because the WHO guideline has stated that if more than one causes are reported on a single line, the first cause should be selected unless there is a cause-and-effect relationship for the reported causes. If the first cause of multiple causes on a single line was the most appropriate as the cause of death, that error would not be considered as a major error. However, if the first cause was selected as the cause of death, the acceptable cause of death had only 3 (23%) cases of 13 cases in this study. Therefore, the error with more than one cause of death on a single line was classified as major error in this study.

â–¶ The two paragraphs on page 3 after Table 1 are a repetition of the detailed definition of major and minor errors in Table 1. It is suggested that in order to prevent duplication of definitions, the two paragraphs after the table should be merged with the table definitions and only the table should be preserved and the next two paragraphs should be deleted. 

→ Thank you for your suggestion. We think it is needs to be explained in detail in the main text. Table 1 was made to make it easier to see contents described in the text at a glance. In the main text, errors that might be difficult to understand were explained with examples. Only definitions are listed in Table 1.

Results: 

The results section is written correctly and logically. Therefore, I have no comment on this section.

Discussion 

The authors discuss the findings of the study favorably. I have no comment in this section. 

Conclusion

â–¶ In the conclusion section, it is suggested that the authors express the determinant factors of the occurrence of major errors in a sentence. 

→ Thank you for your suggestion. We have described the difference in the risk of occurrence of major errors in death certificate based on the definition of UCOD and characteristics of cancer that are comparable to those of other diseases as shown below:

When cancer was listed as the UCOD, there was a lower risk of major errors occurring on the death certificate compared to when the UCOD was cardiovascular disease, cerebrovascular disease, digestive disease, respiratory disease, external causes, or other diseases. Cancer is confirmed in patients by radiologic tests and biopsies during hospitalization and the disease code is registered.

→ We think the clarity of a diagnosis is a key factor considering that there is lower risk of listing unacceptable COD. We have added the above consideration in the conclusion section as follows.

considering the clarity of a diagnosis and accessibility of medical records.

References:

â–¶ References need to be revised to include essential fields such as doi. 

→ We have rechecked all references format. However DOI was not an essential field in the reference format of Healthcare journal.

Reviewer 2 Report

Below are some issues that I found while reviewing the article.

This sentence is incorrect: " The certifier should not select some conditions for entry and reject others". Not all conditions should be included, but only those relevant for the death of that person

This sentence is not correct: "When death certifi- cates only have minor errors such as format errors, there is no difficulty in determining the cause of death". The death certificate, in the absence of an autopsy, only makes educated guesses regarding the causes of death.

Before talking about minor and major errors in drafting the DC, they should be defined.

Materials and methods: why was the study performed on only 6 months? This does not allow a proper analysis of variational issues caused by the seasons. Why were the residents not included as authors? They were involved significantly in the study. Therefore, I feel this study has authorship related issues, which could be considered scientific misconduct.

Why is cancer a disorder that leads to a decrease in major errors? The arguments of the authors do not seem very relevant. Most major issues are related with procedural/metodological isssues; moreover, even in cancers, an alternate chain of causes of death can be identified.

The authors should detail the covariates used for multivariate analysis.

Author Response

(The authors gave the same response as above.)

Reviewer 3 Report

This is an important study that investigated a validity of death certificates in Korea and are almost acceptable even in the current status.

  1.  (Introduction) Although it is written that studies on death certificates with major errors are limited, are there no previous studies investigating a validity of death certificate in Korea? I think some studies have been conducted also in Korea, and you should cite those studies in Introduction and clarify the novelty of this study.
  2. (Methods) It is better to write more about characteristics of the hospital wherein the study was conducted.
  3. (Results and Discussion) It seems to be OK.

Author Response

Thank you for your valuable time and comments. These comments have improved the quality of our manuscript significantly. We have revised this manuscript according to your comments or suggestions.

Reviewer 3

Comments and Suggestions for Authors

This is an important study that investigated a validity of death certificates in Korea and are almost acceptable even in the current status.

â–¶  (Introduction) Although it is written that studies on death certificates with major errors are limited, are there no previous studies investigating a validity of death certificate in Korea? I think some studies have been conducted also in Korea, and you should cite those studies in Introduction and clarify the novelty of this study.

→We have added the references in the introduction section as follows.

Nevertheless, errors on death certificates are common worldwide. Korea is no exception [2-4].

  1. Park, D.K.; Kim, S.Y.; Kang, J.H.; Han, S.H.; Kim, C.H.; Lee, M.C.; Yoo, T.W.; Huh, B.Y. Errors in death certificates in Korea. J Korean Acad Fam Med 1992, 13, 442-449.
  2. Won, T.Y.; Kang, B.S.; Im, T.H.; Choi, H.J. The Study of Accuracy of Death Statistics. J Korean Soc Emerg Med 2007, 18, 256-262.
  3. Yoon, S.H.; Kim, R.; Lee, C.S. Analysis of Death Certificate Errors of a University Hospital Emergency Room. Korean J Leg Med 2017, 41, 61-66.

â–¶ (Methods) It is better to write more about characteristics of the hospital wherein the study was conducted.

→ We have added the following statement in the Methods section.

The study hospital was the only university training hospital in a metropolitan city with 1.1 million people. Therefore, more severely ill patients were admitted in the study hospital than in other hospitals of the region.

(Results and Discussion) It seems to be OK.

Round 2

Reviewer 2 Report

Yhe authors have corrected the issues we have found in the manuscript. As it is, it may be published, provided another round of grammar/syntax check is performed